# Evaluation of *Aspergillus niger* Six Constitutive Strong Promoters by Fluorescent-Auxotrophic Selection Coupled with Flow Cytometry: A Case for Citric Acid Production

**DOI:** 10.3390/jof8060568

**Published:** 2022-05-26

**Authors:** Yudan Lu, Xiaomei Zheng, Yu Wang, Lihui Zhang, Lixian Wang, Yu Lei, Tongcun Zhang, Ping Zheng, Jibin Sun

**Affiliations:** 1College of Biotechnology, Tianjin University of Science & Technology, Tianjin 300457, China; luyd@tib.cas.cn (Y.L.); zhangtc@tust.edu.cn (T.Z.); 2Tianjin Institute of Industrial Biotechnology, Chinese Academy of Sciences, Tianjin 300308, China; zheng_xm@tib.cas.cn (X.Z.); wang_y@tib.cas.cn (Y.W.); zhang_lihui@126.com (L.Z.); wang_lx@tib.cas.cn (L.W.); lei_y@tib.cas.cn (Y.L.); 3Key Laboratory of Systems Microbial Biotechnology, Chinese Academy of Sciences, Tianjin 300308, China; 4College of Life Sciences, University of Chinese Academy of Sciences, Beijing 100049, China; 5National Technology Innovation Center of Synthetic Biology, Tianjin 300308, China

**Keywords:** *Aspergillus niger*, promoter, fluorescence protein, flow cytometry, citric acid, CRISPR/Cas9

## Abstract

*Aspergillus niger* is an important industrial workhorse for the biomanufacturing of organic acids, proteins, etc. Well-controlled genetic regulatory elements, including promoters, are vital for strain engineering, but available strong promoters for *A. niger* are limited. Herein, to efficiently assess promoters, we developed an accurate and intuitive fluorescent-auxotrophic selection workflow based on *mCherry*, *pyrG*, CRISPR/Cas9 system, and flow cytometry. With this workflow, we characterized six endogenous constitutive promoters in *A. niger*. The endogenous glyceraldehyde-3-phosphate dehydrogenase promoter P*gpdAg* showed a 2.28-fold increase in promoter activity compared with the most frequently used strong promoter P*gpdAd* from *A. nidulans*. Six predicted conserved motifs, including the *gpdA*-box, were verified to be essential for the P*gpdAg* activity. To demonstrate its application, the promoter P*gpdAg* was used for enhancing the expression of citrate exporter *cexA* in a citric acid-producing isolate D353.8. Compared with the *cexA* controlled by P*gpdAd*, the transcription level of the *cexA* gene driven by P*gpdAg* increased by 2.19-fold, which is consistent with the promoter activity assessment. Moreover, following *cexA* overexpression, several genes involved in carbohydrate transport and metabolism were synergically upregulated, resulting in up to a 2.48-fold increase in citric acid titer compared with that of the parent strain. This study provides an intuitive workflow to speed up the quantitative evaluation of *A. niger* promoters and strong constitutive promoters for fungal cell factory construction and strain engineering.

## 1. Introduction

*Aspergillus niger* is an important industrial workhorse, owing to its good fitness for industrial fermentation and unique inherent physiological characters, including its high production capacity, robustness to an extreme acid environment, and powerful secretion efficiency to utilize a wide range of carbon sources [1,2,3]. *A. niger* has been widely exploited as cell factories in the industrial production of organic acids and enzymes, including citric acid, gluconic acid, amylase, and glucoamylase [2,3]. Over 80% of worldwide citric acid is produced by submerged fermentation using *A. niger*, with the global citric acid market of nearly 2.0 million tons [2]. With the rapid development of systems biology and synthetic biology, rational engineering has gradually become the dominant approach to reconstruct *A. niger* cell factories with higher titer, yield, and productivity [2,4,5]. Especially the clustered regularly interspaced short palindromic repeats (CRISPR)/CRISPR-associated (Cas) system-mediated genome-editing techniques [6,7,8] have broken the bottleneck of highly efficient genetic manipulation.

In addition to genetic manipulation toolboxes, genetic regulation elements, especially well-controlled promoters, are crucial for strain engineering by fine-tuning the expression of target genes. For instance, some metabolite inducible promoters have been developed for *Aspergillus*, e.g., glucoamylase promoter (P*glaA*) in *A. niger* [9,10,11], but their regulation is dependent on carbon source availability and metabolism, possibly leading to unstable expression in complicated fermentation conditions. Although the metabolism-independent Tet-on system is induced by Doxycycline in *A. niger* [12,13], it is not economically suitable for industrial production. Hence, considering the requirement for strong and stable gene expression, low production cost, and simple fermentation control in industrial biomanufacturing, constitutive promoters are preferred. Compared to inducible promoters, only a few native constitutive promoters have been characterized in *A. niger*, including protein kinase promoter P*pkiA* [14], alcohol dehydrogenase promoter P*adhA* [11], glutamate dehydrogenase P*gdhA* [11], and multiprotein bridging factor promoter P*mbfA* [15]. However, owing to its relatively high activity, the glyceraldehyde-3-phosphate dehydrogenase promoter from *Aspergillus nidulans* (P*gpdAd*) [16] is by far the most frequently exploited promoter in *A. niger* and other *Aspergilli* spp. [17,18,19,20]. Therefore, alternative strong constitutive promoters are desired for engineering multiple genes for strain improvement.

Compared to well-established methods in bacteria and single-celled eukaryotes, such as *Saccharomyces cerevisiae* [21,22], accurate and efficient promoter evaluation approaches are rarely available for filamentous fungi due to the filamentous physiological features of irregular morphologies, which limits the discovery and evaluation of promoters. Traditionally, some enzymes are applied as reporters for promoter evaluation in filamentous fungi, including luciferase [12,13], β-galactosidase of *E. coli* [16,23,24], and β-glucuronidase of *E. coli* [15,25]. However, such enzyme-based approaches involve many labor-intensive and time-consuming steps for activity detection, for example, releasing target proteins from a mycelial culture and determining dry cell weight for accurate enzymatic activity data normalization. Recently, to avoid the influence of mycelial biomass, a fluorescence distribution for spore population is coming into use for the quantitative evaluation of promoter strength by fluorescence protein-coupled flow cytometry [26,27]. This approach provides a much more rapid screening platform for assessing promoters with desirable activity.

To accurately assess the promoter activity in filamentous fungi, it is worthy to note that the promoter–reporter cassette should be targeted to genomic locus-specific integration as a single copy. Huang et al. reported that two isolates of even the same promoter exhibited significantly different transcription levels of nearly two folds, because of random genetic insertion with various copy numbers and integration sites [28]. Apparently, the existence of a non-homologous end joining (NHEJ) system in filamentous fungi caused the indeterminacy of DNA repair. Multiple insertions frequently occur and have been found to improve gene expression [11]. Even with the same copy, integration at different genome loci also caused a variation in gene expression level, due to the impact of chromosomal structure and transcription complex accessibility [29]. Therefore, more genetic manipulation efforts are required to obtain positive isolates before promoter evaluation. Obviously, it is obligatory to have an efficient and easily detectable selection system for genetic manipulation. Otherwise, intensive efforts and much time have to be spent on confirming genotypes of isolates, as it often happens, including randomly picking colonies from selective transformation plates without any indicator, at least twice for spore purification and genotype verification. These conventional procedures impede the rapid promoter characterization in filamentous fungi.

To address these limitations, based on our previously developed 5S rRNA-CRISPR/Cas9 system for *Aspergillus* [6], we established an accurate and simplified promoter evaluation workflow with an intuitive fluorescent-auxotrophic selection in *A. niger*. In detail, the highly efficient CRISPR/Cas9 system ensures precise integration of a promoter-reporting cassette into a specific locus, to avoid inaccurate promoter activity determination. The fusion of a fluorescence protein with an auxotrophic selection marker was used as donor DNAs, which not only provides a reporter for rapidly and quantitatively reflecting the promoter activity via flow cytometry but also offers a double-checked indicator for conveniently distinguishing positive transformants with fluorescence grown on a selective transformation plate without uracil through fluorescence imaging. Using this workflow, we rapidly evaluated six endogenous constitutive promoters and identified the strongest promoter, a native glyceraldehyde-3-phosphate dehydrogenase promoter of *A. niger* (P*gpdAg*), which was a 2.28-fold increase in activity compared to the most frequently used strong promoter, P*gpdAd* from *A. nidulans*, and its application led to significant improvement of citric acid production.

## 2. Materials and Methods

### 2.1. Strains and Cultivation Conditions

The strains used in this study are listed in Appendix A. *Escherichia coli* DH5α (Transgene, Beijing, China) was used for plasmid construction and cultured at 37 °C in Luria–Bertani broth containing ampicillin (100 μg/mL). The citric acid-producing strain *A. niger* D353.8 (*kusA::hph*, *pyrG::hph*, *hyg^R^*) was stored in the lab [30]. *A. niger* strains were cultivated on defined minimal medium (MM), as reported previously [31], or on complete medium (CM) consisting of MM supplemented with 0.5% yeast extract and 0.1% casamino acids. Then, 1.5% agar was supplemented for plates. When necessary, 10 mM uracil was supplemented in the media for the *pyrG* mutants.

### 2.2. Plasmids Construction

The plasmids, protospacers, and primers used in this study are listed in Appendix A, respectively. To establish a rapid promoter evaluation workflow using fluorescent protein–selection marker fusion, the red fluorescence protein-encoding gene *mCherry* was amplified with the primers mCherry-F and mCherry-R, then assembled into the backbone of pSM-AnpyrG [30] amplified with the primers of pSM-AnpyrG-Frev and pSM-AnpyrG-Rrev, using the ClonExpress^TM^ one-step cloning kit (Vazyme, C113), resulting in pFPSM. To evaluate the promoter strength, seven promoter fragments, including P*gpdAd*, P*gpdAg*, P*enoA*, P*pkiA*, P*citA*, P*mdhA*, P*mbfA*, were amplified with the corresponding primers, shown in Appendix A. Intergenic regions located upstream of the coding sequences of the selected genes were used as promoter sequences, and their lengths are shown in Appendix A. Then, these promoter fragments were cloned to the backbone of pFPSM digested with *Nhe*I, using the ClonExpress^TM^ one-step cloning kit (Vazyme, C113), resulting in pYDD1–pYDD7, respectively. These plasmids were used as the templates for generating the donor DNAs containing the *mCherry-pyrG* fused gene expression cassettes driven by different promoters.

To apply the strong promoter for metabolic engineering, the *cexA* gene was amplified with the primers of cexA-F and cexA-R, then cloned into the backbone of pYDD1 and pYDD2, digested with *Nde*I and *Bgl*II, using the ClonExpress^TM^ one-step cloning kit (Vazyme, C113), resulting in pXMD6 and pXMD7, respectively. The *cexA* expression plasmid harboring the truncated *PgpdAg* promoter was amplified using PgpdAg-775-F and PgpdAg-775-R as primers and pXMD7 as a template, resulting in pXMD8.

To construct the sgRNAs targeting *adgA* as the specific genomic locus, two protospacers were predicted by the sgRNAcas9 software [32] and designed with a minimal off-target possibility, as shown in Appendix A. The targeting sgRNA constructs were built by the digestion of sgRNA expression plasmids psgRNA6.0 [6] with *Bbs*I and ligation with synthetic double-stranded oligonucleotides, agdA-sgRNA1-F/agdA-sgRNA1-R and agdA-sgRNA2-F/agdA-sgRNA2-R, resulting in psgRNA6.18 and psgRNA6.19. The linear sgRNA targeting expression cassettes using the 5S rRNA promoter for *A. niger* transformation were amplified with the primers M13F and M13R, using psgRNA6.18 and psgRNA6.19 as the template, respectively, as previously described [6].

### 2.3. A. Niger Strains Construction

The standard protocol of *A. niger* genome editing using the CRISPR/Cas9 system-based 5S rRNA was performed as previously described [6]. For promoter strength evaluation, donor DNA constructs containing *mCherry-pyrG* fused gene expression cassettes driven by different constitutive promoters were integrated into the genome locus of α-glucosidase (*agdA*), to avoid the influence of the genomic context, resulting in the reported strains YDD1–YDD7, respectively. Briefly, 2 μg of the donor DNA constructs was co-transformed into the protoplasts of *A. niger* D353.8 together with the sgRNA targeting constructs AgdA-sgRNA1, AgdA-sgRNA1, and pCas9-AnpyrG. The transformants with fluorescence were selected after detecting under a fluorescence image system (Tanon, Tianjin, China). After subculturing in 24-well plates, genomic DNA of selected transformants was extracted and verified via diagnostic PCR and sequencing analysis with the primers of agdA-g-F and mCherry-R. Copy number analysis was conducted with quantitative PCR (qPCR) Lightcycler 96 (Roche) using ChamQ Universal SYBR qPCR Master Mix (Vazyme) according to manufacturer’s instructions. The qPCR signal of *mcherry* was normalized to the *gpdA* gene as reference. The qPCR primers are listed in Appendix A. The correct single-integration isolates were chosen for further fluorescence detection.

Similarly, citric acid-producing engineered strains XMD6–XMD8 (Appendix A) were also constructed using the same tactics. The only differences were the donor DNAs containing the *cexA* expression cassette, which was amplified with the primers of MH-agdA-sgRNA1-F and MH-agdA-sgRNA1-R, using pXMD6–pXMD8 as the templates, respectively. After subculturing in 24-well plates, genomic DNA of selected transformants was extracted and verified via diagnostic PCR and sequencing analysis with the primers of agdA-g-F and cex-R.

### 2.4. Fluorescence Microscopic Analysis

To detect the fluorescence protein expression strength driven by different promoters, the conidia, hyphae, and mycelial pellets of each sample were prepared for the fluorescence microscopic analysis. Briefly, the conidia were collected with the 0.9% NaCl containing 0.05% Tween-80, after being sporulated on CM plates for 5 days. The hyphae were prepared according to the previous study [33]. Two disinfected coverslips were placed onto the bottom of a small petri dish containing 5 mL of liquid MM with 0.003% yeast extract. After inoculation with 10^6^ spores for 8 h at 30 °C, coverslips with adherent hyphae were placed upside down on an object slide for the microscope analysis. The mycelial pellets were harvested for the analysis after being incubated in the CM liquid media for 48 h. Differential interference contrast (DIC) and red fluorescent images of the cells were captured with a 40× objective using a Leica DM5000B laser scanning confocal microscope (Leica, Wentzler, Germany) with excitation at 543 nm and detection at 586–670 nm. The results were assembled in Adobe Photoshop 7.0 (Adobe, San Jose, CA, USA).

### 2.5. Flow Cytometry Analysis

For quantitative determination of the fluorescence, the flow cytometry was conducted using conidia. *A. niger* strains were first sporulated on CM plates for 5 days. Then, conidia were collected, diluted in PBS, and filtered through a four-layer lens-cleaning paper prior to the flow cytometry analysis using a MoFlo™ XDP cell sorter (Beckman Coulter Inc., Brea, CA, USA). The green (561 nm) laser and 610 nm filter were used for the mCherry fluorescence measurements. The forward scatter (FSC) voltage and mCherry voltage were set as 43 and 610, respectively. Minor gating was performed on data to exclude obvious errors, such as dust particles, cell clusters, and conidia aggregates. A total of 100,000 cells of each sample were recorded and used for mCherry fluorescence and forward scatter (FSC). Flow cytometry results were analyzed with MoFlo XDP Summit 5.2 software (Beckman Coulter Inc., Brea, CA, USA). The *t*-test was used for the data statistical analysis.

### 2.6. Citric Acid Fermentation and Detection

To determine the application of the strongest promoter identified in this study, citric acid production was demonstrated by enhancing the citric acid synthetic flux and efflux. Citric acid fermentation was carried out using the liquefied corn media, as described in a previous study [34]. The final concentration of 1 × 10^5^ spores/mL was inoculated in 20 mL of liquefied corn media in 100 mL shake flasks at 34 °C and 220 rpm for 120 h. The weight of the shake flasks was measured before and after the citric acid fermentation to eliminate measurement errors caused by evaporation. For the citric acid production in the 5 L bioreactor with a stirring paddle device, the same fermentation parameters [34] were utilized for 144 h, but the aeration rate was coupled to the dissolved oxygen concentration (>40%).

For the citric acid detection, supernatants were filtered from cultures using filter paper. Total acids were first titrated using 0.1429 M NaOH with 20 μL 0.1% phenolphthalein as a pH indicator. Next, the supernatants were diluted in sterile distilled water depending on the estimated total acid. Samples were boiled for 15 min at 100 °C, after which supernatants were centrifuged at 12,000 rpm for 5 min and filtered through a 0.22 μm sterile filter membrane. Extracellular organic acids were detected by Prominence UFLC equipped with a UV detector (Shimadzu, Kyoto, Japan) and a Bio-Rad Aminex HPX-87H column (300 × 7.8 mm), according to the procedure described previously [34].

For the mycelial biomass after citric acid fermentation, the cultures were treated as described in a previous study [30]. Briefly, the mycelial cultures were vacuum filtered through filter paper, washed in 5-fold sterile water, and added to pre-weighed falcon tubes. The biomass was incubated at 50 °C until dry (minimum of 24 h), after which the dry weight was determined.

### 2.7. RNA Sequencing and Transcriptomics Analysis

The samples for transcriptomic profiling were prepared and detected according to a previous approach [35]. Briefly, the mycelial samples were collected by rapid vacuum filtration after cultivating for 72 h in a 5 L bioreactor, washed with 100 mL cold sterilized water, and then immediately placed into liquid nitrogen and stored at −80 °C. Total RNAs were extracted using RNAprep pure Plant Kit (DP432, Tiangen, Beijing, China) and assessed using the RNA Nano 6000 Assay Kit of the Agilent Bioanalyzer 2100 system (Agilent Technologies, CA, USA). RNA sequencing libraries were prepared using NEBNext UltraTM RNA Library Prep Kit for Illumina (New England Biolabs, Ipswich, MA, USA) and then sequenced on an Illumina platform. Paired-end reads were generated at Beijing Biomarker technology Co., Ltd. (Beijing, China). Average clean reads of 52.22, 50.02, 51.43, 50.14, and 47.96 million were generated for assembly and further analysis after filtering and trimming of the raw data (Appendix A). The percentage of bases with Phred scores at the Q30 level (an error probability of 0.1%) ranged from 94.30% to 94.65% and the GC content was 53.99–54.28%. These clean reads were then mapped to the reference genome sequence of the progenitor strain, which was annotated according to the reference genome of *A. niger* CBS 513.88 (Accession: PRJNA19263). Among all the samples, 90.06–95.71% of the clean reads were mapped to the reference genome (Appendix A). These data are provided in the supplementary materials. RPKM (reads per kilobase of transcript per million fragments mapped) was applied to measure the expression level of each gene by StringTie using a maximum flow algorithm. Differential expression analysis was performed by edgeR [36], with the criteria for differentially expressed genes of fold change (FC) ≥ 1.5 and *p* value < 0.05. COG (cluster of orthologous groups of proteins) orthologous classification of DEGs was performed on the BMKCloud platform using a COG database (http://www.ncbi.nlm.nih.gov/COG accessed on1 March 2021) [37].

## 3. Results

### 3.1. Fluorescent-Auxotrophic Double Selection Coupling with Flow Cytometry Workflow

To increase the efficiency of promoter activity detection and improve the accuracy of clone selection in *A. niger*, we developed a fluorescent-auxotrophic double selection coupling with CRISPR/Cas9 system and flow cytometry (Figure 1A). The fluorescent protein *mCherry* was fused to the selection marker *AnpyrG* from *A. nidulans*. The promoter P*gpdAd* from *A. nidulans* was used as an example to establish this workflow (Figure 1B). Then, the CRISPR/Cas9 genome editing system was used to integrate this fluorescent-auxotrophic selection cassette at the *agdA* locus. With mCherry-PyrG as double indicators, a simple fluorescence image analysis was used to directly select potential positive transformants from the selective transformation plates without uracil (Appendix A); 30 of 38 primary transformants showed significant fluorescence after incubating for 4 days. Of these, 16 transformants with fluorescence were randomly picked up and reconfirmed by diagnostic PCR (Appendix A) and qPCR; all these isolates were confirmed with the expected genotype. Among them, the isolate YDD1.13 with a single copy of *mCherry-pyrG* fusion inserted at the *agdA* locus was selected for flow cytometry analysis. As shown in Figure 1B and Appendix A, all the conidia and mycelia pellets of *A. niger* YDD1.13 showed distinct red fluorescence, compared to the parent strain D353.8 without fluorescence (Figure 1C). To quantitively measure the promoter’s strength, a population of 100,000 conidia of YDD1.13 was analyzed by flow cytometry. It was observed that YDD1.13 displayed more conidia with high fluorescence intensity (55.16%), compared to that of the parent strain D353.8 (0.27%) (Figure 1D).

### 3.2. Evaluation of Endogenous Constitutive Promoters Using the Fluorescent-Auxotrophic Selection Workflow

To seek strong promoters, we selected six native promoters involved in the central metabolism in *A. niger* and assessed their strength with the above-established method. The native promoters included glyceraldehyde-3-phosphate dehydrogenase promoter (P*gpdAg*), enolase promoter (P*enoA*), pyruvate kinase promoter (P*pkiA*) [38], citrate synthase promoter (P*citA*) [39], malate dehydrogenase promoter (P*mdhA*) [15], and constitutive transcription factor (P*mbfA*) [15]. To avoid the influence of an integration site, all the promoter-reporting cassettes were targeted to the *adgA* gene locus, owing to its good transcription complex accessibility [29]. For each construct, eight transformants with fluorescence were randomly picked up and reconfirmed by diagnostic PCR (Appendix A) and qPCR. It demonstrated that all these selected isolates were confirmed with a single copy of *mCherry-pyrG* fusion inserted at the *agdA* locus. Among them, four positive isolates were selected for flow cytometry analysis. As shown in Figure 2A and Appendix A, the fluorescence intensities of *mCherry-pyrG* fusion constructs driven by six promoters in conidia, hypha from spore germination, and mycelial pellets were significantly higher than that of the parent strain D353.8. It indicated that all the selected promoters could initiate the transcription of *mCherry-pyrG* in different developmental stages, suggesting these promoters were constitutive promoters. The strongest fluorescence of conidia-expressing *mCherry-pyrG* was found to be under the control of P*gpdAg*, whereas the five remaining endogenous constitutive promoters were still lower than P*gpdAd* from *A. nidulans*. The strength order of these promoters was P*gpdAg >* P*gpdAd* > P*pkiA* > P*mdhA* > P*enoA* ≈ P*mbfA* > P*citA* (Figure 2B,C and Appendix A). From the quantitative analysis, P*gpdAg* of *A. niger* showed a 2.28-fold increase in the median fluorescence value compared with P*gpdAd* of *A. nidulans* (Appendix A).

### 3.3. Characterization of Essential Elements in the PgpdAg Promoter

To determine the essential elements of the P*gpdAg* promoter, we conducted the multiple sequence alignment of the P*gpdAg* promoters of six *Aspergilli spp.* and analyzed the truncation of P*gpdAg* in *A. niger* (Figure 3). As shown in Figure 3A and Table 1, six conserved motifs (Motifs I–VI) were predicted at the upstream of −631 to −272 of P*gpdAg* in *A. niger*. Among them, Motif III was identified as *gpdA*-box with two bases different from the *gpdA*-box of P*gpdAd* [16], which comprised a consensus sequence of 5′-CCARATATCGTGMSTCTCCTGCTTTGCCCGGTGTATGAAACCGGAAARG-3′ and located at −521 to −473 in P*gpdAg* from *A. niger* (Table 1). A G/C-rich conserved motif (Motif IV, GCGGCGCDMYCGGGAA) was found at the downstream of *gpdA*-box. Moreover, a conserved motif (Motif VI) was also predicted at the upstream of the transcription start. To unveil the effect of these motifs on promoter activity, the truncated P*gpdAg* strains were constructed with different donor DNAs by the CRISPR/Cas9 system. These donor DNAs were amplified with different forward primers targeting the P*gpdAg* promoter (such as MH-agdA-sgRNA1-F-*PgpdAg*1011) and the common reverse primer MH-agdA-sgRNA2-R (Appendix A). For each construct, four correct isolates with a single copy of the reporter cassette were selected for flow cytometry analysis. As shown in Figure 3B, the impact of the truncated promoters on the transcription efficiency was investigated by the normalized mCherry fluorescence intensity of the verified transformants. Compared with the full-length promoter of 1889 bp, the deletion of −1889 to −1011 (P*gpdAg*-1011) and to −775 (P*gpdAg*-775) caused a slight decrease in fluorescence intensity. However, the removal of Motif I and Motif II (P*gpdAg*-531) led to a significant reduction in fluorescence intensity to 80.13%. Moreover, a further truncation to −319 (P*gpdAg*-319) drastically decreased the fluorescence intensity compared to P*gpdAg*-531, suggesting that key cis-elements including the *gpdA*-box exist. When all the conserved motifs were removed, it was hard to detect any fluorescence, namely, it showed no detectable differences between P*gpdAg*-157 and the negative control D353.8. These data indicated that the promoter region including motifs I–VI is important for full activity of the P*gpdAg* promoter in *A. niger*.

### 3.4. Application of PgpdAg Dramatically Improved Citric Acid Production in A. Niger

To assess industrial application of the endogenous P*gpdAg* promoter, we chose a citrate exporter *cexA* as an engineering target for citric acid production. The donor DNAs with *cexA* controlled by P*gpdAd*, P*gpdAg* and its truncated mutant P*gpdAg*-775 were integrated at the *agdA* gene locus, respectively (Figure 4A). After verified by the genomic PCR diagnosis and sequencing analysis of the four selected transformants (Appendix A), two isolates of each construction were selected for citric acid fermentation in shake flasks. As shown in Figure 4, all *cexA* overexpressed strains significantly increased citric acid titers. The citric acid production in XMD7.1 (74.82 ± 0.35 g/L) and XMD7.2 (75.26 ± 2.79 g/L) with P*gpdAg* was higher than those of XMD6.1 (52.18 ± 0.53 g/L) and XMD6.2 (54.17 ± 0.47 g/L) with P*gpdAd* (Figure 4B), which achieved an increase of 43.37% and 44.22%, respectively. Moreover, after the normalization of the citric acid titers to the biomass of these isolates, it also confirmed this superiority (Figure 4C), suggesting that the biomass of all isolates has no significant difference with the progenitor control strain D353.8. Together with the former promoter evaluation result, our study proved that the activity of the endogenous promoter P*gpdAg* was higher than P*gpdAd* in *A. niger*. Furthermore, *cexA* overexpressed strains under the control of the truncated promoter P*gpdAg*-775 XMD8.1 (75.66 ± 0.33 g/L) and XMD8.2 (74.93 ± 1.09 g/L) also showed a similar citric acid titer with the full-length promoter P*gpdAg* (Figure 4B). These data supported that P*gpdAg* and its truncated mutant P*gpdAg*-775 are able to give much stronger expression of the targets than the commonly used promoter P*gpdAd*.

Due to the high performance of the engineered *A. niger* strains in shake-flask fermentation, we conducted citric acid fermentations of the isolates XMD6.1 and XMD7.1 in 5 L bioreactors. As shown in Figure 5A, compared to XMD6.1, XMD7.1-expressing P*gpdAg*:*cexA* showed an increased citric acid accumulation. In detail, the citric acid titer of XMD7.1 reached up to 114 g/L, which was improved by 1.31-fold and 2.45-fold compared with those of XMD6.1 (87.0 g/L) and D353.8 (46.5 g/L), respectively. The average citric acid productivity of XMD7.1 was 0.79 g/L/h, while that of XMD6.1 was 0.61 g/L/h. These results further confirmed that, in comparison with P*gpdAd*, the endogenous promoter P*gpdAg* displayed a better effect on citric acid production in *A. niger*.

In order to further elucidate the transcription profile of the strains with *cexA* overexpressed by different promoters, comparative transcriptome analysis was conducted using the samples taken from D353.8, XMD6.1, and XMD7.1 in submerged citric acid fermentation for 72 h (Figure 5A). Compared to the parent strain D353.8, the *cexA* transcription level of XMD6.1 and XMD7.1 increased by 2.79-fold and 6.12-fold, respectively (Figure 5B). In comparison to XMD6.1, the *cexA* expression driven by P*gpdAg* in XMD7.1 was improved by 2.19-fold (Figure 5B), which was consistent with the promoter evaluation data using *mCherry-pyrG* as reporter. Moreover, it was observed that, compared to XMD6.1, 736 upregulated genes and 666 downregulated genes were detected in XMD7.1 (Appendix A). These differentially expressed genes were greatly enriched in carbohydrate transport and metabolism (G) (Appendix A), including several upregulated genes in starch hydrolysis, glucose uptake, and glycolysis, such as *amyA* (An12g06930), *glaA* (An03g06550), *mstC* (An02g03540), *mstG* (An05g01290), *mstH* (An15g03940), *pfkB* (An07g02100), and *ppcA* (An11g02550) (Table 2). As to intracellular citrate transportation, mitochondrial citrate/malate carrier protein CtpA (An11g11230) was also synergistically upregulated (Table 2). Meanwhile, with regards to citric acid degradation and by-product oxalate biosynthesis, cytoplasmic ATP-citrate lyase (AcsA, An11g00510 and AcsB, An11g00530), mitochondrial cis-aconitase AcoA (An08g10530), and cytoplasmic oxaloacetate acetylhydrolase OahA (An10g00820) were dramatically downregulated (Table 2).

## 4. Discussion

The filamentous fungus *A. niger* is an important workhorse in industrial biotechnology. Well-characterized promoters in regard to gene expression strength and regulation pattern are essential for strain engineering. However, because of the lack of fast and efficient promoter evaluation approaches, engineering of filamentous fungi is still short of available strong promoters. The filamentous physiological features of hypha polar growth and irregular morphologies bring several technical challenges for development of an accurate and simple approach for promoter characterization. Herein, we established a fast and easily handy fluorescent-auxotrophic selection coupled with flow cytometry workflow (Figure 1) in combination with a CRISPR/Cas9 genetic manipulation system.

In filamentous fungi, to ensure the accuracy of the promoter assessment, the reporter gene should be inserted into the identical, expected integration site as a single copy. However, the NHEJ DNA repair pathway dominates in filamentous fungi [40], and the entailing end processing of the breaks by NHEJ is typically error prone during genetic manipulation [41]. The NHEJ pathway inclines to generate unexpected multi-copy insertions at unspecific sites thus, leading to the inaccuracy of promoter evaluation. To address the integration indeterminacy of reporter genes, we utilized a highly efficient 5S rRNA promoter-driven CRISPR/Cas9 system [6] in an NHEJ-deficient strain [30], which enhanced the efficiency of the precise editing of the reporter genes [6] (Figure 1). To simplify the laborious screening procedure and avoid false-positive clones, we established a fluorescent-auxotrophic selection workflow. PyrG was used as the first selection marker to allow the growth of positive clones on the selective plates without uracil, whereas mCherry provided a second fluorescence checking for the intuitive selection of correct clones. All picked transformants with distinct fluorescence were verified to be correct isolates with the expected genotype (Appendix A). It should be noted that a few transformants did not show as distinct a fluorescence as others, possibly because of the inconsistent growth (Appendix A). However, the transformants with distinct fluorescence were sufficient for screening the desirable isolates.

In contrast to bacteria and single-celled eukaryotes [21,22], the heterogeneous cultures with complex pelleted and dispersed morphologies made it impossible to measure biomass by cell turbidity. However, determining cell dry weight is usually tedious and time consuming [15,16,25]. To eliminate the influence of cell biomass variance and simplify this promoter strength detection procedure, the fluorescence value distribution of conidia populations determined by flow cytometry was applied in this study to reflect the promoter activity (Figure 1). Consistent with the results in *T. reesei* [27,42,43], the performance of the overall 100,000-conidia populations allowed us to quantitatively distinguish the difference of various promoters, overcome the tedious procedures of biomass measurement, and reduce the operating error (Figure 2). Recently, flow cytometry was applied to directly sort germinated conidia [44] or transformed protoplasts without plating [45], demonstrating the potential of high-throughput screening with mCherry-PyrG fusion by fluorescence-activated cell sorting (FACS). Moreover, coupled with droplet-based microfluidics [44], this fluorescent-auxotrophic selection workflow could be applied for screening inducible promoters under various conditions.

Using this workflow, we characterized six endogenous constitutive promoters of *A. niger* and identified a very strong endogenous P*gpdAg* promoter, which is even much stronger than the most frequently used strong promoter P*gpdAd* of *A. nidulans* [16]. In a case application, the *cexA*-expressing construct with P*gpdAg* promoter produced up to 114 g/L of citric acid (Figure 5), which was 2.48-fold higher than the parental strain, suggesting that our study provided an alternative, strong constitutive promoter P*gpdAg*, whereas a previously reported strong promoter P*mbfA* in *A. niger* ATCC1015 [15], whose transcription level was greater than *gpdA* in ATCC1015, showed only 18.01% and 41.01% lower strength than that of P*gpdAg* and P*gpdAd* in our strain D353.8, respectively (Figure 2 and Appendix A). Moreover, the transcription profile of D353.8 suggested a 9.50-fold higher expression of *gpdA* than *mbfA* (Appendix A). Additionally, MbfA was not detected in the intracellular proteome of *A. niger* AB1.13 growing on a defined medium with xylose or maltose as a carbon substrate [46]. This difference of P*mbfA* activity might result from the discrepancy of gene regulation patterns in a different genetic background.

In addition, it is interesting to mention the broad effect of *cexA* overexpression on the transcription profiling (Appendix A and Table 2). Generally, exporters such as CexA are usually focused on their efflux function of transporting metabolites. We observed that CexA overexpression dramatically influenced the transcriptional pattern of genes involved in carbohydrate transport and metabolism (Appendix A), suggesting its impacts on carbon metabolic flux redistribution. Three glucose transporters MstC, MstG, and MstH displayed the similar expression pattern as the *cexA* gene, which was consistent with the previous transcriptional analysis [47]. Additionally, we also discovered that several genes beneficial for intracellular citric acid accumulation were significantly regulated, including starch hydrolysis, precursor supply, mitochondrial citrate/malate shuttle, alternative oxidase respiratory chain, citric acid degradation, and by-product biosynthesis. It is, so far, not clear that such effect is directly caused by the *cexA* gene or by indirect regulation of the intracellular citric acid level.

## 5. Conclusions

In summary, we established an accurate and intuitive fluorescent-auxotrophic selection workflow to efficiently assess promoter activity in *A. niger*, which could pave the way for high-throughput promoter evaluation in filamentous fungi. With this workflow, we showed that P*gpdAg* is the strongest promoter out of the six tested promoters, providing an alternative competent promoter for metabolic engineering in *A. niger*.

## 6. Patents

The engineered promoters described in this paper are covered by patents CN202110268572.8. P.Z., X.Z., J.S., Y.L., W.Z., L.Z., and Y.M. are listed as co-inventors of the patents.

## Figures and Tables

**Figure 1 jof-08-00568-f001:**
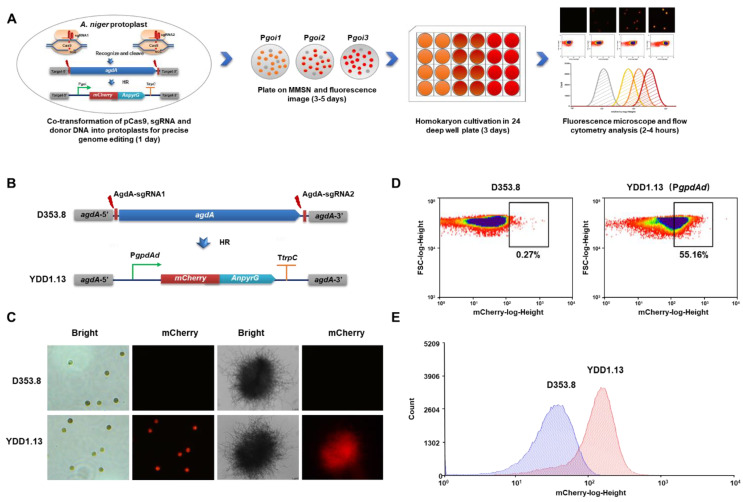
Fluorescent-auxotrophic selection coupled with CRISPR/Cas9 system and flow cytometry. (**A**) Schematic overview of a promoter evaluation workflow based on an intuitive fluorescent-auxotrophic selection. This workflow combined the efficient CRISPR/Cas9 genetic manipulation system, the fluorescent protein-selection marker fused indicator, and flow cytometry-based analysis. CRISPR/Cas9 system was applied for integration of the reporter at the specific genome locus. Then, transformants with distinct fluorescence were picked in 24-deep-well plates based on fluorescence imaging. After genotype verification, fluorescence intensity was detected by laser scanning confocal microscope and flow cytometry. (**B**) Schematic diagram of *mCherry-pyrG-*expressing construct under the control of the P*gpdAd* promoter. The donor DNAs comprised the PgpdAd::*mCherry-pyrG* cassette, and 40-bp micro-homology arms targeted the flanking sequences of *agdA* gene. After being constructed, the donor DNAs were co-transformed with linear sgRNA constructs (*agdA*-sgRNA1 and *agdA*-sgRNA2) and the Cas9 expression cassette into protoplasts of the *kusA* and *pyrG* deficient chassis D353.8. Two DNA double-strand breaks (DSBs) at the flanking sequences of the *agdA* gene were generated by the Cas9 under the guide of two sgRNAs and then were repaired by homologous recombination (HR) with the integration of the donor DNAs, resulting in YDD1.13. (**C**) Representative fluorescence images in conidia and mycelial pellets of YDD1.13-expressing *mCherry-pyrG* with the P*gpdAd* promoter. The parent strain D353.8 was used as negative control. (**D**,**E**) Flow cytometry analysis of the conidia of YDD1.13. The 100,000 spores were analyzed by flow cytometry. Black boxes marking the same value of mCherry-log-height were used for direct comparison of YDD1.13 with the parent strain D353.8.

**Figure 2 jof-08-00568-f002:**
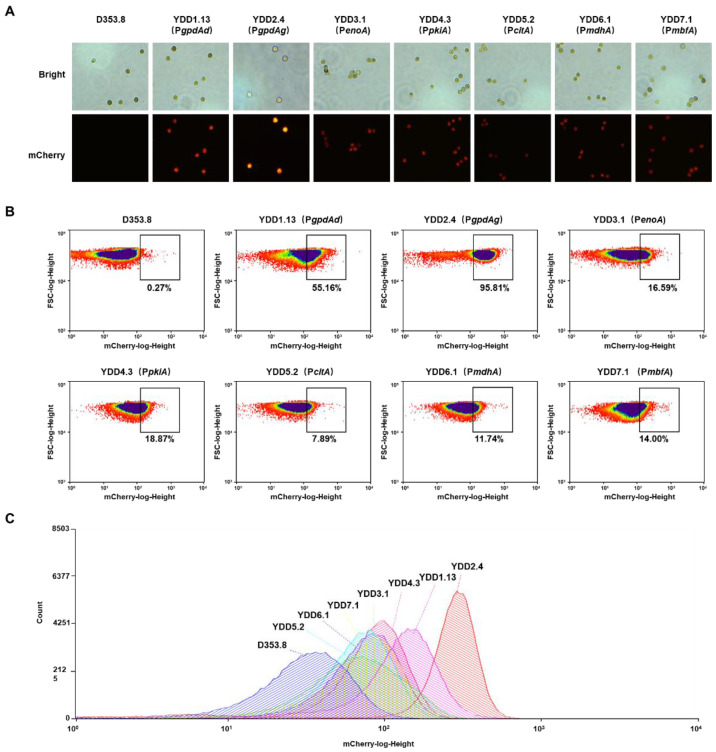
Fluorescence analysis of constructs expressing *mCherry-pyrG* controlled by six constitutive promoters. (**A**) Representative fluorescence images in conidia of constructs expressing *mCherry-pyrG* controlled by six constitutive promoters. The parent strain D353.8 was used as negative control and YDD1.13 was used as positive control. (**B**,**C**) Representative flow cytometry analysis of the conidia of constructs YDD2 to YDD7 expressing *mCherry-pyrG* controlled by six selected promoters. The 100,000 spores were analyzed by flow cytometry. To reduce the interference of background fluorescence from the parent strain D353.8, the black box marking the same value of mCherry-log-height (higher than 10^2^) was used for direct comparison of constructs expressing *mCherry-pyrG* controlled by different promoters. The number below the black box represents the percentage of spores with high fluorescence (mCherry-log-height higher than 10^2^) of each construct. *A. niger* YDD1.13 with the promoter of *PgpdAd* was used as positive control, while the parent strain D353.8 was used as negative control.

**Figure 3 jof-08-00568-f003:**
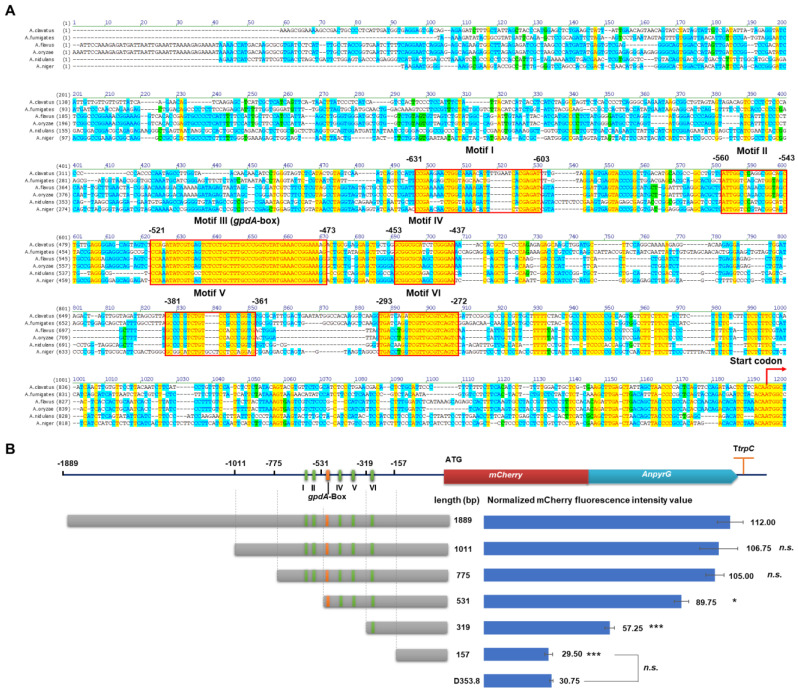
Characterization of essential elements of the P*gpdAg* promoter. (**A**) Multiple sequence alignment of the P*gpdA* promoters of various *Aspergilli spp*. The GenBank accession numbers of the selected *gpdA* genes included *A. niger* CBS 513.88 (An16g01830), *A. nidulans* FGSC A4 (ANIA_08041), *A. clavatus* NRRL1(ACLA_003290), *A. fumigatus* Af293 (AFUA _5G01970), *A. flavus* NRRL 3357 (AFLA_025100), and *A. oryzae* RIB40 (AO090003001322). The conserved motifs are highlighted with red boxes. The transcription start site and start codon are shown as red arrows. (**B**) The conserved motifs’ prediction and truncation test of P*gpdAg* promoter in *A. niger.* GpdA-box and five predicted conserved motifs are represented as orange bars and green bars, respectively. The truncation design of the P*gpdAg* promoter is displayed as gray bars. The mean mCherry fluorescence intensity of each truncation is shown as blue bars. Results are the mean of three replicates, and error bars indicate standard deviations (*n* = 4). Pairwise Student’s *t*-test were conducted between P*gpdAg* truncated mutant relative to the full-length P*gpdAg* reported strain and between the mutant only containing UTR and the parent strain, respectively. The *p* values are indicated as (>0.05, *n.s.*; * < 0.05; *** < 0.001).

**Figure 4 jof-08-00568-f004:**
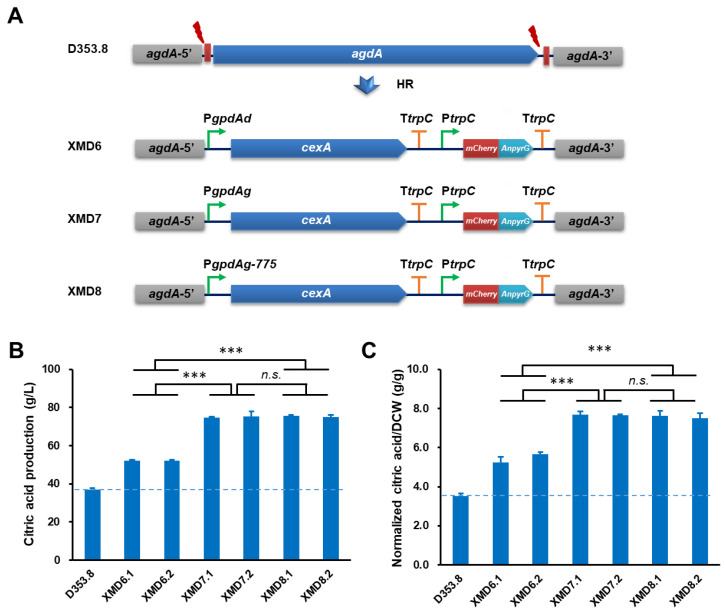
Citric acid production of *A. niger* strains expressing *cexA* under the control of the P*gpdA* promoter. (**A**) Schematic diagram of constructs expressing *cexA* under the control donor DNAs containing the P*gpdA* promoter. The donor DNAs containing P*gpdAd*::*CexA*, P*gpdAg*::*CexA,* and P*gpdAg-775*::*CexA* expressing cassettes were co-transformed with linear sgRNA constructs (agdA-sgRNA1 and agdA-sgRNA2) and a Cas9 expression cassette into the protoplasts of *A. niger* D353.8. Two DNA double-strand breaks (DSBs) at the flanking sequences of *agdA* gene were generated by the Cas9 under the guide of two sgRNAs and then were repaired by HR with the integration of donor DNAs. (**B**,**C**) Citric acid production of the *cexA* overexpressed strains in *A. niger*. Citric acid titer (**B**) and normalized citric acid titer (g citric acid/g dry weight, **C**) were calculated for each strain. Then, 1 × 10^5^ spores/mL were inoculated in 20 mL citrate fermentation media and incubated at 34 °C for 120 h. The extracellular citric acid was determined by the method of HPLC. *A. niger* XMD6.1 with *cexA* overexpression under the control of P*gpdAd* was used as positive control, while the parent strain D353.8 was used as negative control. Results are the mean of three replicates, and error bars indicate standard deviations (*n* = 3). Pairwise Student’s *t-*test was conducted between *cexA* overexpression mutants relative to the parent strains. The *p* values are indicated as (>0.05, *n.s.*; *** < 0.001).

**Figure 5 jof-08-00568-f005:**
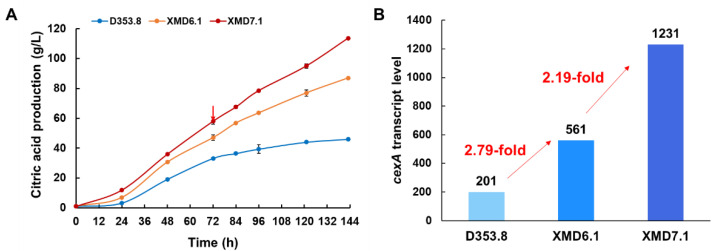
Transcription analysis of *cexA*-expressing constructs in submerged citric acid fermentation. Citric acid production (**A**) and the *cexA* transcript level (**B**) of the *cexA-*expressing constructs in 5 L bioreactors. Then, 1 × 10^5^ spores/mL were inoculated in submerged citric acid fermentation at 34 °C for 144 h. The extracellular citric acid was determined by the method of HPLC. Results are the mean of three replicates, and error bars indicate standard deviations (*n* = 3). The red arrow in (**A**) represents the time point of sampling for RNAseq analysis.

**Table 1 jof-08-00568-t001:** Predicted conserved motifs of P*gpdAg* and their location relative to start codon.

Motif name	Sequence (5′ to 3′) ^1^	Location
Motif I	CCGAacaaCTGGcAaaAcaTtCtCGAGAT	−631, −603
Motif II	ATTGGtCcgtacGGcAgC	−560, −543
Motif III (*gpdA*-box)	CCAaATATCGTGagTCTCCTGCTTTGCCCGGTGTATGAAACCGGAAAgG	−521, −473
Motif IV	GCGGCGCAagcCGGGAA	−453, −437
Motif V	GCggCaTCTGTgcctCCtCCaGGaG	−381, −361
Motif VI	TGAcctGgTCGTTGCGTCAGTC	−293, −272

^1^ The upper cases represent the consensus sequences predicted by multiple sequence alignment, while the lower cases represent the non-conserved bases.

**Table 2 jof-08-00568-t002:** Differentially expressed genes involved in central metabolism in P*gpdA*::*cexA* constructs.

Gene ID	Name	Function	Foldchange		
			XMD6.1 vs. D353.8	XMD7.1 vs. D353.8	XMD7.1 vs. XMD6.1
An17g01710	*cexA*	Citrate exporter	2.79	6.12	2.19
An12g06930	*amyA*	Amylase	2.03	3.76	1.85
An03g06550	*glaA*	Glucoamylase	2.83	6.49	2.29
An02g03540	*mstC*	Low affinity glucose transporter	1.25	1.58	1.26
An05g01290	*mstG*	High affinity glucose transporter	1.25	1.39	1.11
An15g03940	*mstH*	High affinity glucose transporter	2.73	5.75	2.11
An04g06910	*amyR*	Transcription factor	2.35	2.59	1.10
An08g02260	*pgkA*	Phosphoglycerate kinase	1.32	1.37	1.04
An11g02550	*ppcA*	Phosphoenolpyruvate carboxykinase	2.57	4.30	1.67
An07g02100	*pfkB*	6-phosphofructo-2-kinase	1.46	1.90	1.31
An02g12140	*gsdA*	Glucose-6-phosphate dehydrogenase	0.88	0.71	0.80
An02g02930	*rpiB*	Ribose-5-phosphate isomerase	1.35	1.68	1.24
An11g11230	*ctpA*	Citrate/malate carrier protein	1.57	1.26	0.80
An08g10530	*acoA*	cis-aconitase	0.66	0.52	0.79
An11g00510	*acsA*	ATP-citrate lyase	0.45	0.46	1.02
An11g00530	*acsB*	ATP-citrate lyase	0.51	0.50	0.98
An10g00820	*oahA*	Oxaloacetate acetylhydrolase	0.19	0.62	3.23
An15g00070	*mdhAc*	Cytoplasmic malate dehydrogenase	1.26	1.44	1.14
An11g04810	*aox1*	Alternative oxidase	1.37	2.14	1.56
An01g09270	*iclA*	Isocitrate lyase	1.11	5.23	4.72

## Data Availability

The data presented in this study are available in this manuscript and Appendix A and can be requested from the corresponding author.

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
