# Peer review of "Evaluation of Aspergillus niger Six Constitutive Strong Promoters by Fluorescent-Auxotrophic Selection Coupled with Flow Cytometry: A Case for Citric Acid Production"

_jof, 2022, doi:10.3390/jof8060568_

Round 1

Reviewer 1 Report

The paper by Lu et al. describes a new fluorescent-auxotrophic selection method coupled with flow cytometry that can serve as a tool for promoter strength characterization. The authors tested six endogenous promoters with this method and use one of them to test the expression of a gene involved in citric acid production, thus demonstrating its application. I find the work overall solid and rigorous. However, the novelty and advantages of the method are not necessarily clear to me. For example, the finding that the endogenous gpdA promoter works better than the heterologous homolog from A. nidulans is not very surprising. On top, the used CRISPR method has been published by the group before (so is per se not novel) and the “double selection” system is not really double in my point of view (see below). I also have some issues with some of the wording, which tends to overstate the results. I therefore think that there are some points to be addressed and/or clarified and would like to refer to the point-by-point discussion below:

General remarks:

  • A native speaker should correct the paper (especially discussion part)
  • Interpretations should mainly be kept in the discussion part (e.g. l. 271-273; l. 315-317; 440-442)

Major points:

My major criticism of the study lies with the intensive usage of the words “efficient” and “double selection”, which are also part of the title. First: given the amount of verification that is necessary to identify useful transformants is still laborious and time consuming and not much sped up with the presented method (including diagnostic PCRs and qPCRs etc…), the overall method is not really any more efficient than previous methods. The wording should be limited to the specific process of promoter strength analysis by FACS (and not the selection). Regarding “double selection”, I think that also this wording is overstating, since the fluorescence marker and the auxotrophic complementation by pyrG are actually fused and thus a single unit. To my understanding, a real double selection would need these markers to be separate, for example on each ends of the gene of interest, to make sure that it is integrated fulllength. Given these considerations, I suggest to rephrase the title, for example to: “Coupled fluorescent-auxotrophic selection of six Aspergillus niger promoters for strength analysis by flow cytometry: a case for citric acid production” (or the like). Also, the usage of these terms throughout the manuscript should be revised.

A very general question is regarding the six selected promoters (e.g. p.7, l. 297-302): On what basis were the promoters chosen? How were the promoter’s lengths (base pairs) selected and defined? How do you know when a promoter starts (and ends)?

Connecting the method and the chosen promoters, I was wondering how the authors or the method can make sure that the promoters that are characterized are indeed constitutive? The manuscript states that they are, but this is never verified. Also, method-wise FACS is actually limited to the conidia or early germling stage, which is a very specific developmental stage. So every result regarding promoter strength does not necessarily have to be the same in the mycelium stage and older colonies / fermenter pellets etc.. This limitation should be made clear and discussed sufficiently.

In general, the discussion as-is is very result-limited and –oriented. Please, explain the importance and (possible) applications of the method better in the discussion section.

An example:

  1. 498: “[…] surprisingly showed a significantly lower strength […] (Figure 2)”. Do you have an explanation, why this was the case? Please discuss in more detail. Additionally you are describing significant differences, but you are not showing any significance in Figure 2. Which significance test has been performed? Where is it shown?
  2. 354-355: This sentence is not correct, because – for example - you did not make a deletion of a region including only Motif I. Please, replace it with “These data indicate that the promoter region including motifs I-VI is important for full activity of the PgpdAg promoter in A. niger”. Also remove “essential” in the title of 3.3. This could have been argued only if the motifs would have been deleted individually and in the background of the full promoter, not by truncations.

Minor points:

Language:

  1. 52: crucial
  2. Line 57: Doxycycline in
  3. 75: for example
  4. 76: enzymic? activity (enzymatic)
  5. 81: it is
  6. 82: targeted to
  7. 83-84: “even” is repeated
  8. 87: “are common occurred”: please, replace it with “frequently occur”
  9. 92: selection system?
  10. Line 196: The forward scatter (FSC) voltage and mCherry voltage was set as 43 and 610, respectively.
  11. 311: language “promoters was” à “were”, please correct; l. 464 change “dominants” to “dominates”; Do not use abbreviations like “it´s” in an article (l. 481)
  12. 451: Please let the sentence start with “The filamentous fungus…”
  13. 324: Please correct “blank box”
  14. 373: Please, delete the last dot (misprint).

General comments:

Please, explain the strains each time you mention them or state something to make reading easier.

  1. 238 & 243: Content of table S4 does not fit to the content of the text. Maybe you mean table S5?
  2. 258: It should be explained here why the cassette is integrated at the agdA locus (better than in the Material and Methods section).
  3. 306: In table S4 the control strain is missing but was mentioned in the main text.
  4. 415: Please, delete “carbon consumption”. That is not shown in figure 5A.
  5. 469: “[…] which led to significantly enhance […] (Figure 1)” How have you tested that the enhancement is significant? And also in l. 475, l. 478, l. 516
  6. 491: It would be important to mention that FACS was previously used by Beneyton et al. 2016 to sort germinated conidia of Aspergillus niger.

l.524-525: my suggestion would be to modify to: “With this workflow, we showed that PgpdAg is the strongest promoter out of the six tested promoters…”

Figures and tables:

Could you increase the figure quality? Some are pixelated (Fig. 1 A) or too small to be read (e. g. Figure S4, here you cannot read the legend showing the COG classifications)

Figure 1A: should it actually be “homokaryon” cultivation (instead of “homozygote”), and what is “flow cytometry quantitate detection”?

Figure 2: A better description of how the “black box” was set would be helpful. The setting are very important for the interpretation of the results. At least specify in the Methods section.

Figure 2A: Please, indicate what the numbers in the figure caption mean. Figures should be self-explanatory.

Figure 3A: The letters are very small. Better, show schematic representations of the proteins, indicating the important information.

Repetitive: Figure 4A in the main text is the same as Figure S3A

Figure 5: please indicate in the legend what the arrow is indicating (I guess time point of sampling for RNAseq)

Figures S1, S2 and S3: please, indicate what NC means.

Table 2: (l.450) Why is the fold change of “XMD7.1 vs XMD6.1” shown twice and why are the values different, when the same thing is shown? I guess there is a typo here.

Table 2: gene names should be lowercase letters (as in lines 433-434).

Table 2: Please, change in the title “different” by “differentially”.

Table S1 “Description” could be changed into “Genotype”

Author Response

General comment: The paper by Lu et al. describes a new fluorescent-auxotrophic selection method coupled with flow cytometry that can serve as a tool for promoter strength characterization. The authors tested six endogenous promoters with this method and use one of them to test the expression of a gene involved in citric acid production, thus demonstrating its application. I find the work overall solid and rigorous. However, the novelty and advantages of the method are not necessarily clear to me. For example, the finding that the endogenous gpdA promoter works better than the heterologous homolog from A. nidulans is not very surprising. On top, the used CRISPR method has been published by the group before (so is per se not novel) and the “double selection” system is not really double in my point of view (see below). I also have some issues with some of the wording, which tends to overstate the results. I therefore think that there are some points to be addressed and/or clarified and would like to refer to the point-by-point discussion below:

 Answer: We are grateful for all your constructive comments. We have carefully revised the manuscript according to your valuable suggestion. The responses to the comments are listed as follows.

General remarks:

  • A native speaker should correct the paper (especially discussion part)

 Answer: Thanks for your helpful suggestion. We have carefully revised the manuscript with regards to grammar, spelling, and syntax, according to the suggestion from the native speaker.

  • Interpretations should mainly be kept in the discussion part (g. l. 271-273; l. 315-317; 440-442)

 Answer: Thanks a lot. According to this suggestion, we have deleted these corresponding texts in the result section and updated the related texts in the discussion section in the revised manuscript (L271-272; L318; L443; L499-500; L513-514).

L499-500: “suggesting that our study provided an alternative strong constitutive promoter PgpdAg. “

L513-514: “suggesting its impacts on carbon metabolic flux redistribution. “

Major points:

 Comment 1: My major criticism of the study lies with the intensive usage of the words “efficient” and “double selection”, which are also part of the title. First: given the amount of verification that is necessary to identify useful transformants is still laborious and time consuming and not much sped up with the presented method (including diagnostic PCRs and qPCRs etc…), the overall method is not really any more efficient than previous methods. The wording should be limited to the specific process of promoter strength analysis by FACS (and not the selection). Regarding “double selection”, I think that also this wording is overstating, since the fluorescence marker and the auxotrophic complementation by pyrG are actually fused and thus a single unit. To my understanding, a real double selection would need these markers to be separate, for example on each end of the gene of interest, to make sure that it is integrated full-length. Given these considerations, I suggest to rephrase the title, for example to: “Coupled fluorescent-auxotrophic selection of six Aspergillus niger promoters for strength analysis by flow cytometry: a case for citric acid production” (or the like). Also, the usage of these terms throughout the manuscript should be revised.

 Answer: Many thanks for this helpful discussion. To avoid the misunderstanding, we have rephrased the title as “Characterization of Aspergillus niger strong promoters by fluorescent-auxotrophic selection coupled with flow cytometry: a case for citric acid production” and rewritten the related texts in the revised manuscript (L20; L100; L258; L275-276; L295; L461; L472; L493; L523).

Comment 2: A very general question is regarding the six selected promoters (e.g. p.7, l. 297-302): On what basis were the promoters chosen? How were the promoter’s lengths (base pairs) selected and defined? How do you know when a promoter starts (and ends)?

Answer: Many thanks for this helpful discussion. Because genes involved in central metabolism usually exhibit high and continuous expression, we selected five potential constitutive promoters of genes involved in glycolysis and tricarboxylic acid cycle. Additionally, one strong constitutive promoter PmbfA was also selected, which was reported to be even stronger than PgpdAd from A. nidulans (Blumhoff et. al., 2013). Intergenic regions located upstream of the coding sequences of the selected genes were used as promoter sequences. As suggested, we have supplemented these promoters’ length in Table S4 and updated the corresponding texts in the revised manuscript (L132-133).

L132-133: “Intergenic regions located upstream of the coding sequences of the selected genes were used as promoter sequences and their length was shown in Table S4.”

Comment 3: Connecting the method and the chosen promoters, I was wondering how the authors or the method can make sure that the promoters that are characterized are indeed constitutive? The manuscript states that they are, but this is never verified. Also, method-wise FACS is actually limited to the conidia or early germling stage, which is a very specific developmental stage. So, every result regarding promoter strength does not necessarily have to be the same in the mycelium stage and older colonies / fermenter pellets etc. This limitation should be made clear and discussed sufficiently.

 Answer: Many thanks for this helpful discussion. Generally, promoters could be selected according to the gene expression pattern based on transcriptomic data. For instance, Blumhoff et. al., (2013) reported that the most stably expressed genes were determined using the software NormFinder (Andersen et al. 2004), which could be used for screening potential constitutive promoters. Moreover, several promoters of genes involved in central metabolism, such as glyceraldehyde-3-phosphate dehydrogenase promoter of A. nidulans or malate dehydrogenase of A. niger, have been recognized as constitutive promoters (Punt et al., 1990; Blumhoff et. al., 2013). Based on the previous knowledge, we selected five potential constitutive promoters of genes involved in glycolysis and tricarboxylic acid cycle. To further verify this assumption, we have added fluorescence images in hypha and mycelial pellets of constructs YDD2 to YDD7 expressing mCherry-pyrG controlled by six selected promoters in Figure S2C and updated the corresponding texts in the revised manuscript (L297; L308-312).

Figure S2. Constructs expressing mCherry-pyrG controlled by six constitutive promoters.

L297: “To seek for strong promoters, we selected six native promoters”

L308-312: “As shown in Figure 2A, Figure S2C and Table S4, the fluorescence intensities of mCherry-pyrG fusion constructs driven by six promoters in conidia, hypha and mycelial pellets were significantly higher than that of the parent strain D353.8. It indicated that all selected promoters enable to initiate the transcription of mCherry-pyrG in different growth phases, suggesting these promoters are constitutive promoters.

Comment 4: In general, the discussion as-is is very result-limited and –oriented. Please, explain the importance and (possible) applications of the method better in the discussion section.

Answer: Thanks a lot for your constructive suggestion. We have updated the related texts to highlight the potential applications of our workflow in the revised manuscript (L489-494).

L489-494: “Recently, flow cytometry has been applied to directly sort germinated conidia [44] or transformed protoplasts without plating [45], demonstrating the potential of high-throughput screening with mCherry-PyrG fusion by fluorescence-activated cell sorting (FACS). Moreover, coupled with droplet-based microfluidics [44], this fluorescent-auxotrophic selection workflow could be applied for inducible promotes screening under various conditions.”

An example:

  1. 498: “[…] surprisingly showed a significantly lower strength […] (Figure 2)”. Do you have an explanation, why this was the case? Please discuss in more detail. Additionally, you are describing significant differences, but you are not showing any significance in Figure 2. Which significance test has been performed? Where is it shown?

Answer: Thanks for your rigorous comments. As suggested, we have supplemented the foldchange of fluorescence intensity and Pairwise Student’s t-test analysis in Table S4. The fluorescence intensity of YDD7.1 with PmbfA was only 41% of that of PgpdAd from A. nidulans in A. niger D353.8. This difference of PmbfA activity might result from the discrepancy of gene regulation patterns in different genetic back-ground. Correspondingly, we have updated the related text in the revised manuscript (L502-507).

L502-508: “showed only 18.01% and 41.01% lower strength of that of PgpdAg and PgpdAd in our strain D353.8, respectively (Figure 2 and Table S4). Besides, the transcription profile of D353.8 suggested a 9.50-fold higher expression of gpdA than mbfA (Figure S5). Additionally, MbfA was not detected in the intracellular proteome of A. niger AB1.13 growing on defined medium with xylose or maltose as carbon substrate [46]. This difference of PmbfA activity might result from the discrepancy of gene regulation patterns in different genetic background.”

  1. 354-355: This sentence is not correct, because – for example - you did not make a deletion of a region including only Motif I. Please, replace it with “These data indicate that the promoter region including motifs I-VI is important for full activity of the PgpdAg promoter in A. niger”. Also remove “essential” in the title of 3.3. This could have been argued only if the motifs would have been deleted individually and in the background of the full promoter, not by truncations.

Answer: Thanks a lot for your constructive suggestion. To avoid misunderstanding, we have rewritten this sentence in the revised manuscript (L357-359).

L357-359: “These data indicate that the promoter region including motifs I-VI is important for full activity of the PgpdAg promoter in A. niger.” 

Minor points:

 Language:

  1. 52: crucial
  2. Line 57: Doxycycline in
  3. 75: for example
  4. 76: enzymic? activity (enzymatic)
  5. 81: it is
  6. 82: targeted to
  7. 83-84: “even” is repeated
  8. 87: “are common occurred”: please, replace it with “frequently occur”
  9. 92: selection system?
  10. Line 196: The forward scatter (FSC) voltage and mCherry voltage was set as 43 and 610, respectively.
  11. 311: language “promoters was” à “were”, please correct; l. 464 change “dominants” to “dominates”; Do not use abbreviations like “it´s” in an article (l. 481)
  12. 451: Please let the sentence start with “The filamentous fungus…”
  13. 324: Please correct “blank box”
  14. 373: Please, delete the last dot (misprint).

 Answer: Many thanks for these constructive comments. As suggested, we have carefully corrected the corresponding text in the revised manuscript using the “Track Changes” function.

General comments:

Please, explain the strains each time you mention them or state something to make reading easier.

  1. 238 & 243: Content of table S4 does not fit to the content of the text. Maybe you mean table S5?

Answer: Thanks a lot. L245: Table S4 should be Table S5. We have corrected this typo in the revised manuscript (L243-244).

L243-244: “90.06-95.71% of the clean reads were mapped to the reference genome (Table S5).” 

  1. 258: It should be explained here why the cassette is integrated to the agdA locus (better than in the Material and Methods section).

Answer: Thanks a lot. As suggested, we have supplemented the explanation of this integration site in the revised manuscript (L302-304).

L302-304: “To avoid the influence of integration site, all the promoter reporting cassettes were targeted to the adgA gene locus, owing to its good transcription complex accessibility [29].” 

  1. 306: In table S4 the control strain is missing but was mentioned in the main text.

Answer: Thanks a lot. As suggested, we have updated Table S4 in the revised supplementary file.

  1. 415: Please, delete “carbon consumption”. That is not shown in figure 5A.

Answer: Thanks a lot. As suggested, we have modified the related text in the revised manuscript (L419).

L419: “showed an increased citric acid accumulation” 

  1. 469: “[…] which led to significantly enhance […] (Figure 1)” How have you tested that the enhancement is significant? And also in l. 475, l. 478, l. 516

Answer: Thanks a lot. As suggested, to be more rigorous, we have modified the related text in the revised manuscript (L470; L475-476; L478-479).

L470: “which enhanced the efficiency of precise editing of reporter genes [6]” 

L475-476: “All picked transformants with distinct fluorescence” 

L478-479: “the transformants with distinct fluorescence” 

  1. 491: It would be important to mention that FACS was previously used by Beneyton et al. 2016 to sort germinated conidia of Aspergillus niger.

Answer: Thanks a lot for this recommendation. As suggested, we have supplemented this important reference and updated the related text in the revised manuscript (L489-494).

 L489-494: “Recently, flow cytometry has been applied to directly sort germinated conidia [44] or transformed protoplasts without plating [45], demonstrating the potential of high-throughput screening with mCherry-PyrG fusion by fluorescence-activated cell sorting (FACS). Moreover, coupled with droplet-based microfluidics [44], this fluorescent-auxotrophic selection workflow could be applied for inducible promotes screening under various conditions.”

  1. 524-525: my suggestion would be to modify to: “With this workflow, we showed that PgpdAg is the strongest promoter out of the six tested promoters…”

Answer: Thanks a lot. As suggested, we have modified the related text in the revised manuscript (L525-526).

L525-526: “With this workflow, we showed that PgpdAg is the strongest promoter out of the six tested promoters” 

Figures and tables:

Could you increase the figure quality? Some are pixelated (Fig. 1 A) or too small to be read (e. g. Figure S4, here you cannot read the legend showing the COG classifications)

Answer: Many thanks for this helpful suggestion. To improve the figure quality, we have redesigned the Figure 1A and Figure S4 in the revised manuscript.

Figure 1A: should it actually be “homokaryon” cultivation (instead of “homozygote”), and what is “flow cytometry quantitate detection”?

Answer: Thanks a lot. According to this helpful suggestion, we have revised the texts in Figure 1: the “homozygote” has been replaced by “homokaryon” and “flow cytometry quantitate detection” has been replaced by “flow cytometry analysis”.

Figure 2: A better description of how the “black box” was set would be helpful. The setting is very important for the interpretation of the results. At least specify in the Methods section.

Answer: Thanks a lot. According to this helpful suggestion, we have supplemented the description of how to set “black box” in the legend of Figure 2 in the revised manuscript (L325-328).

L325-328: “To reduce the interference of background fluorescence from the parent strain D353.8, the black box marking the same value of mCherry-log-Height (higher than 102) was used for direct comparison of constructs expressing mCherry-pyrG controlled by different promoters.”

Figure 2A: Please, indicate what the numbers in the figure caption mean. Figures should be self-explanatory.

Answer: Thanks a lot. According to this helpful suggestion, we have supplemented the description of numbers in the legend of Figure 2 in the revised manuscript (L328-329).

L328-329: “The number below the black box represent the percentage of spores with high fluorescence (mCherry-log-Height higher than 102) of each construct.”

Figure 3A: The letters are very small. Better, show schematic representations of the proteins, indicating the important information.

Answer: Thanks a lot for this discussion. However, we respectfully suggest that Figure 3A showed the multiple nucleotide sequence alignment of the PgpdA promoters of various Aspergilli spp. Thus, there is no schematic representations of the proteins. If it means the sequences of each Motif, the letters in schematic representations of Motifs were larger for easy reading in Figure 3A.

Repetitive: Figure 4A in the main text is the same as Figure S3A

Answer: According to this helpful suggestion, we have deleted the Figure S3A and its related legend.

Figure 5: please indicate in the legend what the arrow is indicating (I guess time point of sampling for RNAseq)

Answer: Thanks a lot for this helpful suggestion. We have supplemented the description of red arrow in the legend of Figure 5 in the revised manuscript (L449-450).

L449-450: “The red arrow in (A) represented the time-point of sampling for RNAseq analysis.”

Figures S1, S2 and S3: please, indicate what NC means.

Answer: Many thanks for this helpful suggestion. We have supplemented the description of NC in the legends of Figure S1-3.

“The parent strain D353.8 was used as negative control, which was represented as “NC” in each electrophoretogram.”  

Table 2: (l.450) Why is the fold change of “XMD7.1 vs XMD6.1” shown twice and why are the values different, when the same thing is shown? I guess there is a typo here.

Answer: Thanks a lot for this helpful suggestion. The “XMD7.1 vs XMD6.1” in the fourth column should be “XMD7.1 vs D353.8”. We have corrected this typo in the revised manuscript (L452).

Table 2: gene names should be lowercase letters (as in lines 433-434).

Answer: Thanks a lot for this helpful suggestion. We have corrected the gene names in the revised Table 2 (L452).

Table 2: Please, change in the title “different” by “differentially”.

Answer: Thanks a lot for this helpful suggestion. We have corrected the title of Table 2 in the revised manuscript (L455).

Table S1: “Description” could be changed into “Genotype”.

Answer: Many thanks for this helpful suggestion. We have changed “Description” into “Genotype” in Table S1.

Reviewer 2 Report

The submitted manuscript represents novel and original work regarding the citric acid production by a filamentous fungi. The manuscript is very well written and presented, with interesting results and a very concise discussion section. 

This reviewer congratulates the team, and recommend acceptance in the current form.

Author Response

Comments: The submitted manuscript represents novel and original work regarding the citric acid production by a filamentous fungus. The manuscript is very well written and presented, with interesting results and a very concise discussion section. 

This reviewer congratulates the team, and recommend acceptance in the current form.

Answer: We are very grateful for your encouraging comments. Thanks a lot.

Round 2

Reviewer 1 Report

Review of revised manuscript by Lu et al. “Characterization of Aspergillus niger strong promoters by fluorescent-auxotrophic selection coupled with flow cytometry: a case for citric acid production”

The paper by Lu et al. has improved significantly by the revision, but some issues still remain, which are detailed below.

General remarks:

I asked for a native speaker to correct the paper (especially discussion part), which the authors claim has happened, but several mistakes and typos remain. E.g. line 59: “doxycycline”, line 91: “Multiple insertions frequently occur and have been found to improve gene expression“, line 499: should be “In contrast to bacteria,…”, or line 515: should be “…inducible promoter screening…”. Please revise the manuscript again.

Title:

I also suggested to rephrase the title to: “Coupled fluorescent-auxotrophic selection of six Aspergillus niger promoters for strength analysis by flow cytometry: a case for citric acid production”. The title was rephrased but still sounds like a series of promoters was characterized, which is clearly misleading, since only six were tested for strength and only one was (slightly) characterized by truncation constructs. I therefore think that the title should be modified again – and still suggest my original title idea. If the authors would like to emphasize the usability of their setup as a tool, then this can be done, but would also require rephrasing.

Selection of the promoters:

The authors added information regarding the promoters they used for the study (e.g. Table S4), but it is still unclear for example on which basis the very individual promoter lengths were chosen. Was this based on previous studies? Then please add the relevant citations to Table S4 for example. If this was based on some other criteria, then please explain them in more detail.

Also, in Table S4, it should be indicated from which stage the mCherry data were recorded from.

Fig. S2c: this was added/modified to include more developmental stages to strengthen the argument of constitutive expression. This is fine per se, but I would like to ask to 1) change “hypha” to “hypha from germlings” and 2) change “growth phases” to “developmental stages” (lines 318-320 in the track-changes version).

Author Response

Manuscript ID: jof-1706358

Evaluation of Aspergillus niger six constitutive strong promoters by fluorescent-auxotrophic selection coupled with flow cytometry: a case for citric acid production

Dear Dr. Caroline Song and Dr. Silvia Alexandra Avram,

We are submitting a revised version of the manuscript ID jof-1706358 entitled “Evaluation of Aspergillus niger six constitutive strong promoters by fluorescent-auxotrophic selection coupled with flow cytometry: a case for citric acid production”. We are very grateful to you and the reviewers for the valuable comments and the constructive suggestions, which helps to improve the manuscript. We carefully considered all the comments and rewritten the corresponding texts. All revisions have been clearly marked in the revised manuscript. Point-by-point responses to the comments are listed as follows. We look forward to hearing from you regarding this revised manuscript.

With best regards.

Ping Zheng

To Reviewer 1:

Review of revised manuscript by Lu et al. “Characterization of Aspergillus niger strong promoters by fluorescent-auxotrophic selection coupled with flow cytometry: a case for citric acid production”

General comment: The paper by Lu et al. has improved significantly by the revision, but some issues still remain, which are detailed below.

 Answer: We are very grateful for all your constructive comments. We have carefully revised the manuscript according to your valuable suggestion. The responses to the comments are listed as follows.

General remarks:

I asked for a native speaker to correct the paper (especially discussion part), which the authors claim has happened, but several mistakes and typos remain. E.g. line 59: “doxycycline”, line 91: “Multiple insertions frequently occur and have been found to improve gene expression”, line 499: should be “In contrast to bacteria,…”, or line 515: should be “…inducible promoter screening…”. Please revise the manuscript again.

 Answer: We apologized for these mistakes and typos still in the revised manuscript. Many thanks for your careful revision. As suggested, we have modified the corresponding texts in the revised manuscript (L57; L87; L480; L493). Moreover, we have also carefully checked and corrected some undetected space errors in the track-changes version (L300-302).

L57: “induced by Doxycycline in A. niger

L87: “Multiple insertions frequently occur and have been found to improve gene expression”

L480: “In contrast to bacteria”

L494: “screening inducible promoters”

L300-302: “pyruvate kinase promoter (PpkiA) [38], citrate synthase promoter (PcitA) [39], malate dehydrogenase promoter (PmdhA) [15], and constitutive transcription factor (PmbfA) [15].”

Comment 1: Title: I also suggested to rephrase the title to: “Coupled fluorescent-auxotrophic selection of six Aspergillus niger promoters for strength analysis by flow cytometry: a case for citric acid production”. The title was rephrased but still sounds like a series of promoters was characterized, which is clearly misleading, since only six were tested for strength and only one was (slightly) characterized by truncation constructs. I therefore think that the title should be modified again – and still suggest my original title idea. If the authors would like to emphasize the usability of their setup as a tool, then this can be done, but would also require rephrasing.

Answer: Many thanks for this helpful discussion. To avoid the misunderstanding, we have rephrased the title as “Evaluation of Aspergillus niger six constitutive strong promoters by fluorescent-auxotrophic selection coupled with flow cytometry: a case for citric acid production”.

Comment 2: Selection of the promoters: The authors added information regarding the promoters they used for the study (e.g. Table S4), but it is still unclear for example on which basis the very individual promoter lengths were chosen. Was this based on previous studies? Then please add the relevant citations to Table S4 for example. If this was based on some other criteria, then please explain them in more detail.

Answer: Thanks for your helpful discussion. In the previous studies, 0.9-kb (Dave and Punekar, 2011) 1.0-kb (Storms et al., 2005) or 1.5-kb (Blumhoff et al., 2013) fragments located upstream of the coding sequences of the selected genes were usually used as promoter sequences. According to these previous studies and genome sequences analysis, we also selected the intergenic upstream regions of the selected genes as promoter sequences. As suggested, we have supplemented the corresponding citations in Table S4 and the supplementary file.

Table S4. Promoter strength evaluation by flow cytometry analysis.

Strains

Promoter

Reference

Gene ID

Length (bp)

Function Description

D353.8

/

/

/

/

YDD1.13

PgpdAd

[3]

AN8041

680

Glyceraldehyde-3-phosphate dehydrogenase

YDD2. 4

PgpdAg

\

An16g01830

1889

Glyceraldehyde-3-phosphate dehydrogenase

YDD3.1

 PenoA

\

An18g06250

1056

Enolase

YDD4.3

 PpkiA

[4]

An07g08990

1299

Pyruvate kinase

YDD5.2

 PcitA

[5]

An09g06680

1127

Citrate synthase

YDD6.1

 PmdhA

[6]

An15g00070

1240

Malate dehydrogenase

YDD7.1

 PmbfA

[6]

An02g12390

1122

Transcription coactivator

Reference

  1. Punt, P.J.; Dingemanse, M.A.; Kuyvenhoven, A.; Soede, R.D.; Pouwels, P.H.; van den Hondel, C.A. Functional elements in the promoter region of the Aspergillus nidulans gpdA gene encoding glyceraldehyde-3-phosphate dehydrogenase. Gene. 1990, 93,101-109.
  2. Storms, R.; Zheng, Y.; Li, H.; Sillaots, S.; Martinez-Perez, A.; Tsang, A. Plasmid vectors for protein production, gene expression and molecular manipulations in Aspergillus niger. Plasmid. 2005, 53, 191-204.
  3. Dave, K.; Punekar, N.S. Utility of Aspergillus niger citrate synthase promoter for heterologous expression. J. Biotechnol. 2011, 155,173-177.
  4. Blumhoff, M.; Steiger, M.G.; Marx, H.; Mattanovich, D.; Sauer, M. Six novel constitutive promoters for metabolic engineering of Aspergillus niger. Appl. Microbiol. Biotechnol. 2013, 97,259-267.

Comment 3: Also, in Table S4, it should be indicated from which stage the mCherry data were recorded from.

Answer: Thanks a lot. The mCherry data were recorded from the conidia stage in Table S4. As suggested, we have supplemented the description of the mCherry fluorometric analysis in Table S4.

1 For each construct, the mCherry intensity of 100,000 conidia were analyzed by flow cytometry.”

Comment 4: Fig. S2c: this was added/modified to include more developmental stages to strengthen the argument of constitutive expression. This is fine per se, but I would like to ask to 1) change “hypha” to “hypha from germlings” and 2) change “growth phases” to “developmental stages” (lines 318-320 in the track-changes version).

Answer: Thanks a lot. As suggested, we have modified the corresponding texts in the revised manuscript (L310-312).

L310: “hypha from spore germination”

L312: “in different developmental stages”
